# Description of a New Species and Lectotypification of Two Names in *Impatiens* Sect. *Racemosae* (Balsaminaceae) from China

**DOI:** 10.3390/plants10091812

**Published:** 2021-08-31

**Authors:** Shuai Peng, Peninah Cheptoo Rono, Jia-Xin Yang, Jun-Jie Wang, Guang-Wan Hu, Qing-Feng Wang

**Affiliations:** 1CAS Key Laboratory of Plant Germplasm Enhancement and Specialty Agriculture, Wuhan Botanical Garden, Chinese Academy of Sciences, Wuhan 430074, China; pengshuai183@163.com (S.P.); peninahrono@yahoo.com (P.C.R.); yangjxgz@163.com (J.-X.Y.); 18790123619@163.com (J.-J.W.); qfwang@wbgcas.cn (Q.-F.W.); 2Sino-Africa Joint Research Center, Chinese Academy of Sciences, Wuhan 430074, China; 3Wuhan Botanical Garden, Chinese Academy of Sciences, University of Chinese Academy of Sciences, Beijing 100049, China

**Keywords:** lectotype, new taxa, phylogeny, protologue, Sichuan, taxonomy

## Abstract

*Impatiens longiaristata* (Balsaminaceae), a new species from western Sichuan Province in China, is described and illustrated here based on morphological and molecular data. It is similar to *I. longiloba* and *I. siculifer*, but differs in its lower sepal with a long arista at the apex of the mouth, spur curved downward or circinate, and lower petal that is oblong-elliptic and two times longer than the upper petal. Molecular analysis confirmed its placement in sect. *Racemosae*. Simultaneously, during the inspection of the protologues and type specimens of allied species, it was found that the types of two names from this section were syntypes based on Article 9.6 of the International Code of Nomenclature for algae, fungi, and plants (Shenzhen Code). According to Articles 8.1, 9.3, and Recommendations 9A.1, 9A.2, and 9A.3, the lectotypes of these two names are here designated.

## 1. Introduction

*Impatiens* L. (Balsaminaceae) is one of the most species-rich genera of angiosperms with more than 1000 species [1,2]. It is mainly distributed in tropical and subtropical Africa, Madagascar, southern India and Sri Lanka, eastern Himalayas and Southeast Asia [3,4]. In contrast to the wide distribution of the genus, many *Impatiens* species are endemic to specific mountains or valleys. Most of the species are forest plants, usually growing in moist semi-shaded places, rarely in semi-arid places [2,5]. Flowers of *Impatiens* are zygomorphic and have a wide variety of morphology and colors [4,5]. The diversity of species has resulted in an extremely complex interspecific relationship in this genus. Since Hooker and Thomson [6] provided the first infrageneric classification for *Impatiens* based only on morphological data, several infrageneric classifications of the genus have been proposed in the last century [7,8,9]. However, all of these classifications were based on a few macromorphological characters only. Fortunately, phylogenetic studies using nucleotide sequences have made great contributions to our understanding of the evolutionary relationships of *Impatiens* in the last two decades [4,5,10,11,12,13]. Of these studies, the most notable is that in which Yu et al. [5] proposed a new classification based on phylogenetic and morphological evidence in 2016. They divided this genus into two subgenera: *I.* subgen. *Impatiens* and *I.* subgen. *Clavicarpa* S. X. Yu ex S. X. Yu & Wei Wang. *I.* subgen. *Impatiens* was further divided into seven sections, of which *I.* sect. *Racemosae* Hook. f. et Thomson was characterized by its inflorescences racemose, many-flowered; lateral sepals 2, rarely 4 with inner 2 reduced; capsule linear; and seed ovoid.

China is a hotspot for the distribution of *Impatiens* and has numerous balsam species, most of which are distributed in the southwest of the country. The Flora of China lists 227 species, of which 187 are endemic [14]. According to Du et al. [15], a total of 69 new taxa of *Impatiens* have been described in China between 2000 and 2019, and 13 new taxa [16,17,18,19,20,21,22,23,24,25,26,27,28] and three new records [28,29,30] have been published in the last two years, bringing the total species reported for China to nearly 290. 

In 2019, we collected an *Impatiens* species that was morphologically similar to *Impatiens siculifer* Hook. f. [31] (p. 246) and *I. longiloba* Craib [32] (p. 161), from Shimian county, Sichuan province. The examination of the morphology and molecular analyses suggested that our *Impatiens* was significantly different from the two allied species and other known species. Moreover, while examining the protologue of allied species we noted that two names, *I. aquatilis* Hook. f. [31] (p. 247) and *I. clavicuspis* Hook. ex W. W. Smith [33] (p. 337), were based on syntypes according to Article 9.6 of the International Code of Nomenclature for algae, fungi, and plants (Shenzhen Code) [34]. Hence, in the present study on *Impatiens* from China, a new species is described, and lectotypes are designated for two names.

## 2. Materials and Methods

In this study, the morphological terminology for species description follows Grey–Wilson [2]. The infrageneric classification of *Impatiens* uses the recent research result from Yu et al. [5]. Herbaria are indicated with their acronyms, according to Thiers [35]. 

### 2.1. Identification and Description of New Taxon

During a botanical exploration in western Sichuan in 2019, we noticed an interesting aristate yellow-flowered species of *Impatiens* growing in the moist area under a broad-leaved forest (the arista was up to 1 cm long in the bud stage). Specimens were collected, preserved, and deposited in the Herbarium of Wuhan Botanical Garden, Chinese Academy of Sciences (HIB) for further identification. We then dissected the flower and carefully reviewed related literature [14,15,16,17,18,19,20,21,22,23,24,25,26,27,28,29,30,36,37] and herbaria specimens in a bid to ascertain the identity of the collected species. However, we were unable to match the new plants with any described species based on morphology. Molecular phylogenetic analyses (described below) resolved the new plants as being closely related to *I. longiloba* and *I. siculifer*, both of which differ in morphology. Therefore, we conclude that our plants represent a new species, and we describe it as such below. 

The morphological description of new species was based on careful field observation of plants, measurement of herbarium specimens, as well as plant photographs taken in the field. Comparisons between the new species and allied species were based on photographs of herbarium specimens and literature [14,31,32,36,38].

### 2.2. Taxon Sampling and DNA Sequencing

Based on previous studies [4,5,10], 2 DNA sequences (nrITS and plastid DNA *atpB-rbcL*) of 117 species (containing 3 outgroups) were selected to build the phylogenetic tree. All sequences, except new species, were downloaded from GenBank, and GenBank accession numbers are presented in Appendix A. The corresponding sequence of the new species was provided in this study.

Genomic DNA of the new taxon was extracted from silicagel-dehydrated leaf tissue using Mag-MK Plant Genomic DNA Extraction Kits (Sangon Biotech, Shanghai, China). The primer sequences and polymerase chain reaction (PCR) amplified procedure of nrITS and *atpB-rbcL* sequences followed Yuan et al. [5] and Janssens et al. [10]. PCR products were sequenced by Tsingke Biotechnology Co., Ltd. using sanger sequencing.

### 2.3. Phylogenetic Analyses

All phylogenetic analyses were done under PhyloSuite v1.2.2 [39], an integrated and scalable desktop platform. Sequence data were aligned with MAFFT [40] and concatenated using PhyloSuite. The best evolutionary models for two regions (ITS and *atpB-rbcL*) were selected using ModelFinder [41] according to the AIC criterion. The best-fit models for the Maximum likelihood (ML) analysis are GTR + F + R5 (ITS) and GTR + F + R3 (*atpB-rbcL*), and for Bayesian inference (BI) they are GTR + F + I + G4 (ITS) and GTR + F + G4 (*atpB-rbcL*). Maximum likelihood (ML) analysis was inferred using IQ-TREE [42] for 1000 standard bootstraps. Bayesian inference (BI) analysis was inferred using MrBayes 3.2.7a [43], running for 10,000,000 generations, starting from different random trees and sampled every 1000 generations, in which the initial 25% of sampled data were discarded as burn-in.

### 2.4. Lectotypification

The protologue and relevant literature were investigated as references for the nomenclaturally typical element of *Impatiens aquatilis* and *I. clavicuspis*. It was evident that more than one collection was cited in the protologue, indicating that those specimens are syntypes in light of Article 9.6 from the *International Code of Nomenclature for algae, fungi, and plants* (Shenzhen Code) [34]. The designation of lectotypes was necessary following Articles 8.1, 9.3 and recommendations 9A.1, 9A.2, and 9A.3 of the Shenzhen Code.

## 3. Results and Taxonomic Treatment

### 3.1. New Taxon

*Impatiens longiaristata* S. Peng, G. W. Hu & Q. F. Wang *sp. nov.* (Figure 1A–I, the LSID for the name is: 77219511-1).

Diagnosis: *Impatiens longiaristata* is similar to *I. longiloba* and *I. siculifer*, but can be distinguished by its lower sepal with a long arista (1 cm) appearance from the bud stage, spur curved downward or circinate, and lower petal that is oblong-elliptic, tip rounded, and two times longer than the upper petal.

Type: CHINA. Sichuan province, Shimian County, under a broad-leaved forest, 28°58′13″ N, 102°13′49″ E, elevation 2680 m, 1st September 2019, *Shuai Peng*, *Jia-Xin Yang* & *Jun-Jie Wang PS-0096* (holotype: HIB 0189503!; isotype: HIB 0189504!, 0189505!, 0189506!).

#### 3.1.1. Description 

Perennial herbs, 50–90 cm tall. Stem erect or procumbent in the lower part, succulent, branched, glabrous, lower nodes swollen with some fibrous roots. Leaves alternate; lamina obovate, elliptic or elliptic-lanceolate, 8–13 × 3–6.5 cm, glabrous, base obtuse to attenuate, apex caudate-acuminate; margin coarsely crenate, mucronulate in the depression between crenatures; lateral veins 5–7 pairs; upper leaves sessile, lower leaves petiole up to 2.5 cm in length, base biglandular. Inflorescence in upper leaf axils, pseudo-terminal raceme with 3–5-flowered, peduncles 6.2–8.3 cm long, rachis 3–5 cm long. Flower yellow with red spots. Bracts ovate-lanceolate to linear-lanceolate, ca. 4–5 × 2–3 mm, caducous (visible during the bud stage and caducous during flowering). Pedicels 2–3 cm long, slender, glabrous. Lateral sepals 4 (2 pairs), upper pair yellowish, membranaceous, lanceolate, ca. 5 × 2 mm; lower pair green, ovate-lanceolate, ca. 7 × 3 mm; lower sepal yellow with few red spots, obliquely funnelform, gradually narrowed into a decurved or circinate spur, 1.5–2 cm long, mouth 1–1.5 cm wide, apex of mouth with a long arista ca. 9–12 mm (conspicuous in flower bud stage). Dorsal petal yellow with red spots, shallowly cucullate, suborbicular or broadly obovate, ca. 0.8–1.2 × 0.7–1.0, dorsally glabrate without a crest, middle incrassate, margin membranous and often revolute; lateral united petals ca. 3 cm long, the lower petal of each pair about two times longer than the upper petal; upper petal yellow with red spots in inner margin, triangular semicircle ca. 15 × 7 mm; lower petal yellow with a few red spots at the base, oblong-elliptic, ca. 26 × 7 mm, tip rounded. Stamen filaments linear, slightly swollen above; anthers obtuse.

#### 3.1.2. Etymology

The specific epithet “*longiaristata*” is derived from the lower sepal with a long arista at the apex of mouths. It is most conspicuous when the flowers are in bud.

#### 3.1.3. Phenology

The new species was observed flowering from August to September and fruiting from September to October.

#### 3.1.4. Distribution and Ecology

*Impatiens longiaristata* is currently known only from the type location in Shimian County, Sichuan Province, China. It grows in moist places under broad-leaved forests and frequently in association with *Pilea* sp., *Athyrium* sp., *Salvia* sp., and *Impatiens oxyanthera*.

#### 3.1.5. Phylogenetic Position

In this study, phylogenetic analyses based on combined datasets of nuclear ITS and plastid *atpB-rbcL* intergenic spacer DNA sequences confirmed the new species to be in the *Impatiens* sect. *Racemosae* and a close relative to *I. longiloba* and *I. siculifer* (Figure 2).

#### 3.1.6. Conservation Status

The new species is currently only known from a single population with ca. 300 mature individuals at the type locality. We thereby consider that *I. longiaristata* can be evaluated as vulnerable (D1) based on the International Union for Conservation of Nature (IUCN) Red List Categories and Criteria [44].

#### 3.1.7. Additional Specimens Examined

*Impatiens longiloba*: THAILAND. Doi Kai (Intanon) Doi Angka Pah Ageam, North peak, 18°7.5′ N, 98°6′ W, altitude 2120–2125 m, *Garrett, H.B.G. 72* (type: K, K000675562 & K000675563 digital images!), *Impatiens siculifer*: CHINA. Yunnan, Mengtsz S.E., altitude 5000 ft, *Henry, A. 10038A* (K, K000694601 digital image!), 10038B (K, K000694602 & K000694603 digital images!; PE, 00039613 digital image!). Guizhou, Tou-chan (Dushan), 2 June 1898, *Bodinier, E.M. 2335* (P, P00780759 digital image!; E, E00313594 digital image!).

### 3.2. Lectotypification

Lectotypification is among the key aspects of plant systematics. During the perusal of literature, the typification of two species named *Impatiens aquatilis* and *I. clavicuspis* was considered paramount. This prompted an investigative review of original literature and materials in the author’s herbarium during the collection period. We found that three gatherings of herbarium were cited in protologues of *I. aquatilis* and four gatherings were cited in protologues of *I. clavicuspis*. The digital images of herbaria cited by these two names, holding in K, P and E were studied. Based on Articles 8.1, 9.3, and recommendations 9A.1, 9A.2 and 9A.3 of the *Shenzhen Code* [34], the lectotypification of the two names is achieved here. 

*Impatiens aquatilis* Hook. f. (1908: 247). Lectotype (designated here): China, Yunnan, alt. 5500 ft. *Ducloux, F. 532* (K, K000694599 digital image!).

Hook f. primarily described *Impatiens aquatilis* in 1908 [31], based on three gatherings cited in the original material from Yunnan in China, viz. [*Ducloux, F. 532* (K, K000694599 digital image!)], [*Henry, A. 12,580* (K, K000694600 digital image!; E, E00313628 digital image!)], and [*Henry, A. 9430* (K, K000694598 digital image!; E, E00313627 digital image!)]. A critical examination of digital images from all collections reveals that the *Ducloux, F. 532* digital image registered *Henry, A.* as the collector on the mounted herbarium sheets in Kew. In addition, this collection was cited at the Museum d’Histoire Naturelle herbarium but could not be traced among online digital images under *Ducloux, F. 532* collections in P but was rather found in the Kew digital herbarium. It is not clear whether at one point herbarium specimens were transferred from P to K or the digital images were obtained by the Royal Botanical Gardens, Kew during databasing and digitization sessions of 185,937 specimens at P [45]. This makes *Ducloux, F. 532* gathering significant. Notwithstanding this, *Ducloux, F. 532* has descriptions from the author’s handwriting with a signature of his name and also has clear illustrations with candid footnotes and hand drawings. After a comparison among all syntypes, *Ducloux, F. 532* from Yunnan, Mengtsze alt. 5500 ft. in K is selected to serve as the lectotype (Figure 3) for the name *I. aquatilis* because it is the best representative material. It is also complete, and its descriptions rhyme well with the details of the original publication.

*Impatiens clavicuspis* Hook. ex W. W. Smith (1915: 337). Lectotype (designated here): China, Yunnan, Mengtze, altitude 1676 m. *Henry, A. 9762* (K, K000199723 digital image!).

Smith W.W described *Impatiens clavicuspis*, listing four collections in the protologue [33] from different gatherings, viz. [*Henry, A. 9762*] from Yunnan, Mangtze (K, K000199723 digital image!), [*Ducloux, F. 432*] from Yunnanfu (K, K000199724 digital image!), [*Forrest, G. 1004*] from Yunnan, the divide between the Mekong and Shang valley on Jung-yuch-Talifu road (K, K000694594 digital image!; E, E00313653 digital image!) and [*Forrest, G. 1006*] from Yunnan. Upper Burma, N.W. Yunnan Ming Kwang divide (K, K000694595; E, E00313652). All examined collections of digital images had hand drawings of the plant reproductive illustrations (pods and flower morphological characteristics). Furthermore, *1004* and *1006* had accompanying comprehensive handwritten fieldnotes that matched the original description. However, after a critical examination of all these syntypes, we found the specimen displaying the best representative material and locality information (including elevation), numbered *9762*, best fitted the original portal description and was preserved in the Kew herbarium. Moreover, this specimen retains the original pigmentation, making the collections more appealing and providing a precise representation of the living plant species. Hence, we designate it here as the lectotype (Figure 4).

## 4. Discussion and Conclusions

*Impatiens longiaristata* is morphologically and phylogenetically allied to *I. longiloba* and *I. siculifer*. The information available about the distribution of the latter two species suggests that *I. longiloba* is endemic to the type locality in Thailand and that *I. siculifer* is distributed in southwest China [9,10] and Vietnam [46]. Morphologically, all three have a similar type of inflorescences (racemose), the color of flowers (yellow with red spots), and the shape of lower sepals (funnelform). However, *I. longiaristata* can be easily distinguished from these two allied species by its long arista of the lower sepal, the shape of the spurs, and the lateral united petals (Figure 1). The lower sepal has an arista ca. 1 cm long at the apex of the mouth in *I. longiaristata*, while it is shortly rostellate or aristate (no more than 5 mm) in *I. siculifer* and apiculate in *I. longiloba*. The spur is curved downward or circinate in *I. longiaristata*, while it is curved upward in *I. siculifer* and slightly curved at the tip in *I. longiloba*. The lateral united petals of *I. longiaristata* are ca. 3 cm long, the upper petal is triangular and semi-circular, and the lower petal is oblong-elliptic, tip rounded, ca. 1.8 cm long, and two times longer than the upper petal, whereas in *I. siculifer* the lateral united petals are ca. 1.8 cm long, the upper petal is subtriangular, and the lower petal is lorate with an acuminate tip, and in *I. longiloba* the lateral united petals are ca. 2.5 cm long and the lower petal is lorate, elongate, and four times longer than the upper petal. The lateral veins of leaves are 5–7 pairs in *I. longiaristata*, but they are 10–12 pairs in *I. longiloba* and 5–11 pairs in *I. siculifer*. The lateral sepals are two pairs in *I. longiaristata* and *I. longiloba*, but just a pair in *I. siculifer*. The detailed morphological comparisons between new species and allied species are shown in Table 1. The results of the BI and ML analyses showed that these three species were in a well-supported clade (PP = 0.99 and BP = 85), in which *I. longiaristata* and *I. longiloba* formed a highly supported clade (PP = 1.00 and BP = 100) with *I. siculifer* forming a sister clade. It was also noted that *I. longiaristata* was morphologically closer to *I. longiloba* than to *I. siculifer* (Table 1). 

Based on detailed morphological studies and phylogenetic analysis, *Impatiens longiaristata* is a new taxon in science. Detailed descriptions, color plates, phylogenetic analysis, and comparisons with species of similar taxa are provided. The lectotypes of *I. aquatilis* and *I. clavicuspis* are also designated.

## Figures and Tables

**Figure 1 plants-10-01812-f001:**
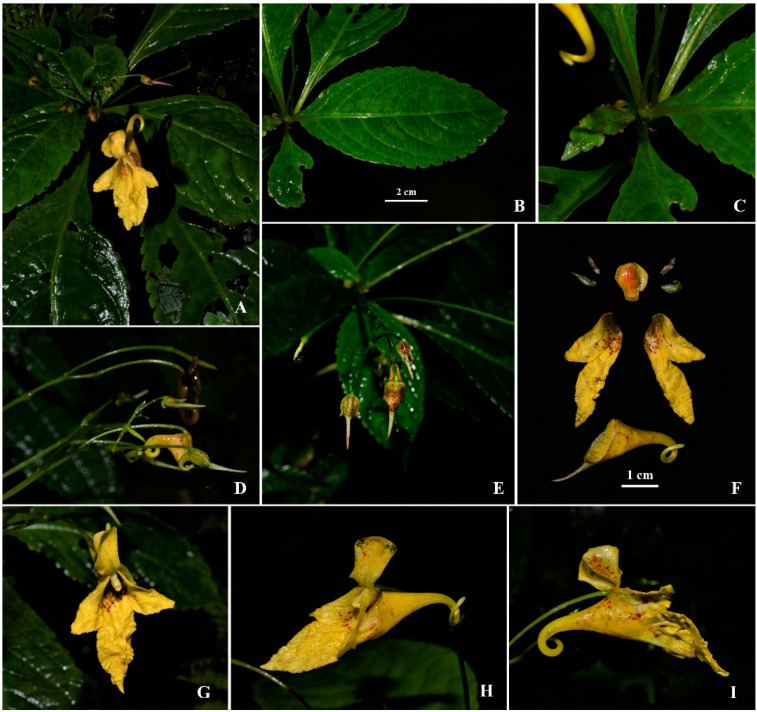
*Impatiens longiaristata* S. Peng, G. W. Hu & Q. F. Wang. (**A**) Habit, (**B**) Leaf, (**C**) Petiole, (**D**) Flower buds, (**E**) Inflorescence, (**F**) Flower anatomy, (**G**) Front view of flowers, (**H**,**I**) Lateral views of flowers (photographed by Shuai Peng).

**Figure 2 plants-10-01812-f002:**
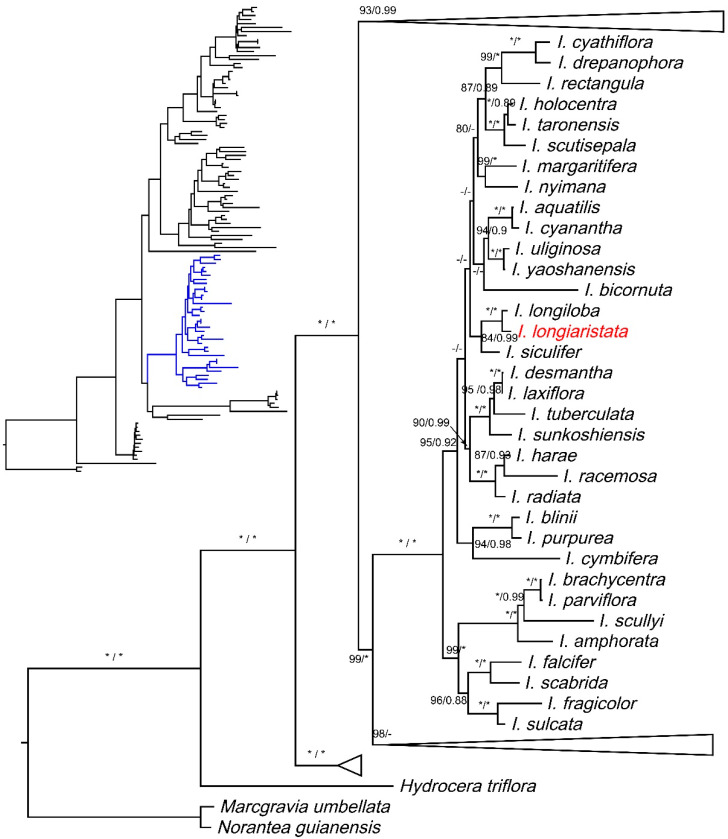
Phylogenetic tree based on a combined dataset of nrITS and plastid *atpB-rbcL* DNA sequences. The bootstrap percentages (BP) for maximum likelihood (ML) and posterior probabilities (PP) of Bayesian inference (BI) are shown in the branch (BP/PP, the dash [-] indicates BP < 80 or PP < 0.80 at a node, and the asterisk [*] indicates BP = 100 or PP = 1.00). Only the ML tree is shown, because its topology is nearly identical to the BI tree. *Impatiens longiaristata* is highlighted in red.

**Figure 3 plants-10-01812-f003:**
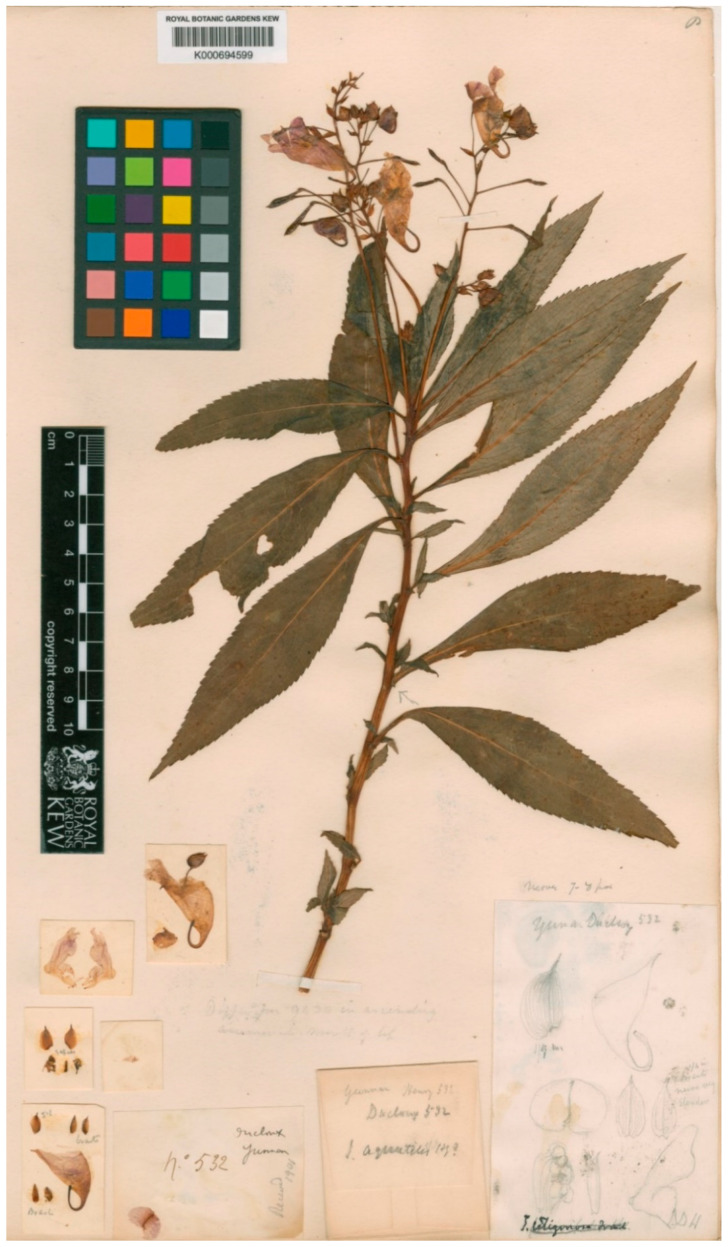
Lectotype of *Impatiens aquatilis* Hook. f. (designated here). Source from http://specimens.kew.org/herbarium/K000694599 (accessed on 6 April 2021). *© copyright of the Board of Trustees of the Royal Botanic Gardens, Kew*.

**Figure 4 plants-10-01812-f004:**
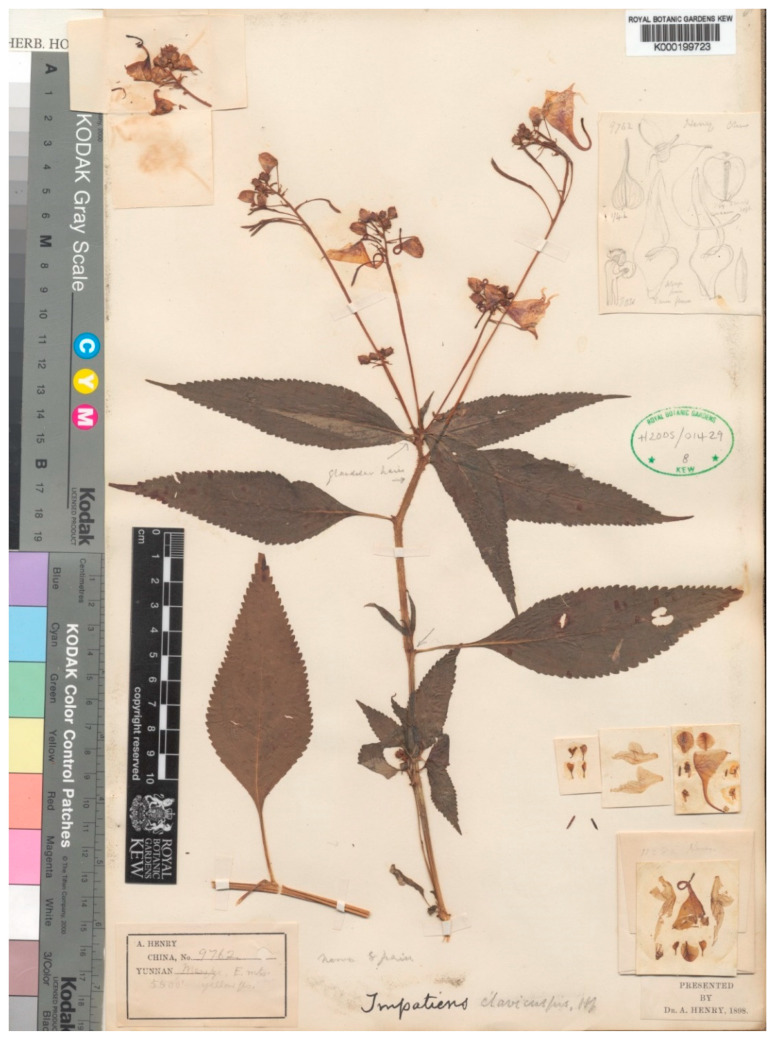
Lectotype of *Impatiens clavicuspis* Hook. ex W. W. Smith (designated here). Source from http://specimens.kew.org/herbarium/K000199723 (accessed on 6 April 2021). *© copyright of the Board of Trustees of the Royal Botanic Gardens, Kew*.

**Table 1 plants-10-01812-t001:** Morphological comparisons of *Impatiens longiaristata*, *I. longiloba*, and *I. siculifer*.

Characters	*I. longiaristata*	*I. longiloba*	*I. siculifer*
Lamina shape	obovate, elliptic, or elliptic-lanceolate	oblong-elliptic	ovate-lanceolate or elliptic-lanceolate
Lamina size	8–13 × 3–6.5 cm	3–11 × 1.2–5 cm	5–13 × 2.5–5 cm
Lateral veins	5–7 pairs	10–12 pairs	5–11 pairs
Length of peduncle	6.2–8.3 cm	3–5 cm	5.5–7 cm
Number of flowers	3–5	2–4	5–8
Bracts	caducous	caducous	persistent
Number of lateral sepals	4 (2 pairs)	4 (2 pairs)	2 (1 pair)
Arista of lower sepal	aristate at apex of mouth, arista ca. 1 cm long in bud and mature flower	shortly apiculate	shortly rostellate or aristate at apex of mouth
Shape of spur	curved downward or circinate	slightly curved at tip	curved upward
Length of lateral united petals	ca. 3 cm long	ca. 2.5 cm long	ca. 1.8 cm long
Shape of lower petal	oblong-elliptic, tip rounded, 2 times longer than the upper petal	lorate, 4 times longer than the upper petal	lorate, tip acuminate

## Data Availability

Not applicable.

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
