# Peer review of "Description of a New Species and Lectotypification of Two Names in Impatiens Sect. Racemosae (Balsaminaceae) from China"

_plants, 2021, doi:10.3390/plants10091812_

Round 1
Reviewer 1 Report
Accept in present form
Author Response
Dear reviewers:
Thank you for your Suggestions and Comments on our manuscript.
Reviewer 2 Report
I really appreciate all the work put into any corrections made by the authors. Now everything looks much, much better.
Author Response

(The authors gave the same response as above.)

Reviewer 3 Report
The manuscript describes a new taxon in Impatiens from China and adds some lectotypification notes about two further species.
A first flaw of the manuscript is that "the ingredients of the dish do not mix well". I strongly suggest to prepare a separate paper about the lectotypifications of the two taxa, otherwise not directly involved in the central issue of the contribution. Several critic points, however, were enlightened about these typifications (see the uploaded file).
As far as the principal issue of the manuscript is concerned, I am not a specialist in Balsaminaceae but I think that the discovery in China of an Impatiens very similar to I. longiloba (endemic to Thailand) is very interesting. Possibly, it is to be regarded as a new species or, at least a new infraspecific taxon (but see below). However, I do not known if such a novelty could justify the publication on Plants. I leave this decision to the Editor.
In alternative, the authors could provide a wider contribution on taxonomy of Chinese Impatiens, for example checking the untypified names in the section and including the description of the new species.
In every case, the paper should provide more information about the habitat and the conservation status of the new population, and also a stronger base to separate it into a new taxon.
The molecular study is probably sufficient to assess the inclusion of the new population into the sect. Racemosae, within the clade with I. siculifer and I. longiloba, but (a) 2 sequences are in general too few for a so wide and differentiated group to discriminate the new taxon, (b) it is not clear how many items per taxon were analyzed, unless reading Tabe S1 (only one, I see), (c) - more important - I cannot see how the phylogram could justify the separation of I. longiaristata from I. longiloba.
As the descriptions regards, I note that the diagnosis is rather vague, and only in the discussion a puntual differentiation toward the allied species is provided. The description itself could be implemented (see the file).
According to the authors, I. longiloba would lack the arista. Neverthless, I read in the protologue of this latter species "ore apiculato".
In general, some redundances occur throughout the manuscript, and also the English language sometimes sounds a little odd to me. Even the structure and organization of the manuscript is to be better defined.
In conclusion, I think that the new population might deserve a taxonomic recognition, but a more robust presentation is needed. Therefore, the manuscript should be re-submitted, after a thorough revision..
Best regards,
Emanuele Del Guacchio
Author Response
Dear reviewers:
Thank you for your Suggestions and Comments on our manuscript. We hereby resubmit our manuscript having addressed the concerns and added in your valuable insights to improve it. Please see the attachment

Round 2
Reviewer 3 Report
Dear Editor,
Dear Authors,
I recommended major revisions to the manuscript in my first review report. I see the text changed only marginally. Therefore, I hope you will understand that I cannot express a favorable opinion about the new version.
In particular, I remain not convinced by the present structure of the contribution.
I think that it is rather odd to include two typifications only because the authors incidentally came across two untypified names. I would add that these typifications not only are not related to the new taxon, but they do not present any particular or intriguing issue for the reader.
On one hand, the authors state that including the nomenclature of all the Chinese taxa of the section (as I suggested) would result as a "too big a job to do all at once" (sic!); on the other hand, they refuse to separate the typifications into a more punctual but more consistent contribution.
However, please note that the following part of the abstract is to be deleted: "it was found that the types of two names from this section were syntypes based on Article 9.6 of the International Code of Nomenclature for algae, fungi, and plants (Shenzhen Code). According to Articles 8.1, 9.3, and Recommendations 9A.1, 9A.2, and 9A.3, lectotypes of these two names are here designated". An example of more concise sentences: "Lectotypifications for two names of the same section, i.e. FIRST NAME and SECOND NAME are proposed. The lectotypes of FIRST NAME and SECOND NAME are syntypes preserved respectively at FIRST HERBARIUM CODE and SECOND HERBARIUM CODE".
I leave this choice to the Editor, but I would to underline the, in its current form, the manuscript does not seem to match the target and the impact of the journal.
The only part improved is the information about habitat and conservation status, but it is honestly too poor and concise.
The description, that I judged as lacunose, has been improved only by a single, almost useless sentence. I asked "Any details about anthers, ovary or seeds?", but no answer came by the authors.
The phylogram can only confirm that the new taxon is to be included into sect. Racemosae, and that it is closely related to I. longiloba (please, pay attention to the English form in the results). Besides, the morphological similarity with I. longiloba can be otherwise deduced by the presence in both the species of the appendage.
I am sorry not to be of further help.
Best regards,
Emanuele Del Guacchio